# Low serum lipase levels in mothers of children with stunted growth indicate the possibility of low calcium absorption during pregnancy: A cross-sectional study in North Sumatra, Indonesia

**Dina Keumala Sari**[1]*, **Rina Amelia**[2], **Dewi Masyithah**[3], **Kraichat Tantrakarnapa**[4]

1 Department of Nutrition, Faculty of Medicine, Universitas Sumatera Utara, Medan, North Sumatra, Indonesia, 2 Department of Public Health, Faculty of Medicine, Universitas Sumatera Utara, Medan, North Sumatra, Indonesia, 3 Department of Parasitology, Faculty of Medicine, Universitas Sumatera Utara, Medan, North Sumatra, Indonesia, 4 Department of Social and Environmental Medicine, Faculty of Tropical Medicine, Mahidol University, Bangkok, Thailand

* dina@usu.ac.id

**Data Availability Statement:** All relevant data are within the paper and its Supporting information files and other files.

## Abstract

Stunting is caused by various factors, including low nutritional intake in the first two years of life. This study aimed to investigate the differences in sociodemographic factors and mineral, vitamin, and enzyme parameters in mothers associated with the occurrence of stunting in children. We conducted a cross-sectional study from September to November 2020 on North Sumatra Island, Indonesia. The data collected included sociodemographic characteristics, pregnancy history, birth history, food intake, and laboratory examinations, including measurements of calcium, iron, zinc, vitamin D, pancreatic amylase, and serum lipase levels. This study included 50 healthy mothers aged 18–50 years old with children aged 2 to 60 months. There was a significant difference in serum calcium levels between the groups of mothers of children with normal and stunted growth ($p = 0.03$, mean difference±standard error (SE) = 0.23±0.12, 95% CI: 0.19–0.45). All of the study subjects were categorized as vitamin D deficient. The mean lipase level in the group of mothers of children with stunted growth was significantly lower than that in the group of mothers of children with normal growth ($p = 0.02$, mean difference±SE = 4.34±1.83, 95% CI: 0.62–8.06). The conclusion was that serum lipase levels were significantly lower in mothers of children with stunted growth compared to mothers of children with normal growth. Serum lipase levels this low are likely to indicate that a mother is unable to meet her child's calcium needs during pregnancy, increasing the child's risk of stunted growth.

## Introduction

The World Health Organization (WHO) defines stunting as the failure of growth and development in children due to insufficient nutritional intake over a long period, recurrent

**Funding:** We would like to thank to The Ministry of Educational and Culture Republic of Indonesia for funded this study. This work was under the World Class University Program of Universitas Sumatera Utara Year 2020. The Grant Number is 1879/UN5.1.R/SK/PPM/2020, and the funding was awarded to DKS.

**Competing interests:** The authors have declared that no competing interests exist.

infectious diseases, and inadequate psychosocial stimulation [1–7]. Stunting still occurs in developing countries, such as India, Ethiopia, and Indonesia, and this is an important concern because it affects the health of individuals in adulthood and impacts the economy of a country [2–4, 8, 9].

Indonesia still has a high number of individuals with nutritional deficiencies in several provinces, including North Sumatra [10–13]. Numerous diseases due to nutritional disorders such as stunting, hydrocephalus, and various others still occur in this region. The stunting rate for 0- to 60-month-old children in North Sumatra in 2019 was 30.1%, down 2.3% percent from that in 2018 [13, 14]. Children with stunted growth are susceptible to disease and have below-average levels of intelligence and low productivity [15–18].

Stunting is caused by various factors, including low nutritional intake in the first two years of life [10]. A mother's nutritional status is very important in determining whether a child's growth is stunted or not [11, 19–21]. The nutritional status of the mother, including the fulfilment of macronutrient and micronutrient needs, will affect the nutritional status of the child, which determines the likelihood of stunting [22–24]. Despite the implementation of programs designed to tackle stunting, cases of stunting are still common. Naturally, this creates the desire to understand the main causes of stunted growth.

The nutritional deficiencies mainly associated with stunted bone growth are calcium and vitamin D deficiencies [25–28]. However, deficiencies of micronutrients, such as minerals and vitamins, are often not detected because patients initially present with subclinical symptoms but will have long-term defects [29–31]. Micronutrient deficiencies are critical, causing disturbances in nutritional status, especially for children, that will last until adulthood [7, 32].

The metabolism of calcium and vitamin D is closely related to their absorption in the body and involves the body's digestive enzymes [33, 34]. Digestive enzymes support the absorption of fat, and sufficient levels of fat support the absorption of calcium and vitamin D, especially in pregnant mothers; this is important because it affects the nutritional status of children [35]. Based on previous research, further study is needed to explore the role of vitamins, minerals, and enzymes in preventing stunting. Research on enzymes that influence nutrient absorption, including pancreatic amylase and lipase, which are involved in carbohydrate absorption and fat absorption, respectively, has not been widely reported. Research on lipase enzymes that influence the absorption of calcium and vitamin D has also not been reported; for this reason, the serum lipase level was one of the variables included in this study.

The existence of various diseases affects the conditions for providing nutritional intake in childhood. Indonesia, which has been affected by the COVID-19 pandemic since February 2020, has also experienced various economic problems that affect the availability of food [36, 37]. Most people rely on food items that are obtained from the area around their residence [38]. Influential micronutrients from the surrounding soil include minerals, which are from metal and inorganic compounds [39–41].

In this study, we evaluated the factors influencing the occurrence of stunting in children, especially during the COVID-19 pandemic. The factors examined included sociodemographic characteristics, pregnancy history, birth history, nutritional intake, and laboratory tests, including measurements of calcium, iron, zinc, vitamin D, pancreatic amylase, and serum lipase levels. This study focused on mothers with children aged 2 to 60 months old. The results of this study may provide new insights regarding the occurrence of stunting so that prevention measures can be implemented. However, in the initial stages of research, the focus was on vitamins, minerals and enzymes that influence a mother's nutritional needs.

## Materials and methods

### Study design

This was a cross-sectional study that included 50 mothers and 50 children. We recorded data on sociodemographic characteristics, maternal pregnancy and birth history, anthropometric characteristics of the mothers and children, daily food intake of the mothers and children, and maternal levels of calcium, iron, zinc, vitamin D, pancreatic amylase, and serum lipase. This study also assessed the soil and water content to examine the condition of food sources from the local area.

The determination of the sample size was based on the research hypothesis, an unpaired two-sided hypothesis with comparison testing of two unpaired groups. This study compared the serum vitamin/mineral and enzyme levels between two groups. To determine the sample size, the number of samples taken with a type I error was set at 5%, and a two-sided hypothesis was used. The Zα value was 1.96. The type II error was set at 20%, with Zβ = 0.84. This study used a significance value of 0.005, and the calculation was carried out using the existing hypothesis and by taking the largest sample size.

This research was conducted from September to November 2020 during the COVID-19 pandemic, but all studies were conducted by implementing the relevant health protocols. Health protocols included cleaning tools with a disinfectant after each use, using personal protective equipment, and maintaining a distance of 1–2 metres.

We studied stunting in North Sumatra because of its known high stunting rate. The districts with high rates of stunting include Medan City, Deli Serdang, Langkat, Simalungun, Dairi, Pakpak Barat, Tapanuli Tengah, Mandailing Natal, Padang Lawas, Padang Lawas Utara, Nias, South Nias, West Nias, North Nias, and Gunung Sitoli. This research was conducted in one location at the Community Health Centre for Simpang Dolok Village, Datuk Lima Puluh District, Batu Bara District, Simalungun and Batubara Border, North Sumatra, Indonesia. This research took place in one district to avoid having heterogeneous data.

This study divided all subjects into two groups based on anthropometric examinations of the children (mothers were grouped according to the nutritional status of their child). The examination was carried out by measuring the height for age and determining the nutritional status of the child based on the WHO growth curve. All study subjects were divided into two groups: a group of children with normal growth and a group of children with stunted growth.

### Participants

The research participants were healthy women aged 18–50 years old with children aged 2 to 60 months. The exclusion criteria were as follows: pregnant women; nursing mothers; women with impaired kidney and liver function, chronic disease or other metabolic disorders; and children with chronic diseases. Confounding factors in this study were maternal gastrointestinal disorders and maternal metabolic disorders, which we controlled for in the inclusion and exclusion criteria. Participants were recruited through appeals from the integrated service centre for mothers and children and the community health centre in the target village at the appointed time, and then research was carried out by recording the number of mothers and children who volunteer to participate. The arrival of mothers and children with stunted or normal growth was not determined, but they were randomly assigned at the time of the study.

At the time of participant selection, 62 research subjects were enrolled, but after applying the exclusion criteria, 50 research subjects were included. All subjects were divided into two groups: a group of children with normal growth (32 mothers and their children) and a group of children with stunted growth (18 mothers and their children).

## Instrument

A questionnaire was utilized for the collection of data for this research to determine the characteristics of the research subjects, including age, occupation type, occupation type of their partner/husband, pregnancy history, birth history, and other information. Data regarding the soil and water content were checked in a laboratory.

Soil and water laboratory analyses were carried out to determine soil fertility, which may affect crops in the area. The quality of vegetables and fruits grown in local soil can also be affected. On soil examination, the magnesium level was 87.2 mg/100 g (normal limit: 25–44 mg/100 g), the potassium level was 66.96 mg/100 g (normal limit: 30–250 mg/100 g), the phosphorus level was 0.251 mg/kg (normal limit: 0.3–0.5 mg/100 kg), and the nitrogen content was 0.1% (normal level: 0.06–0.17%). Magnesium, potassium, and nitrogen levels were within normal limits, and only phosphorus levels were lower than normal [42].

Groundwater examination was carried out, and the pH of the water was 4.49 (normal pH is 6.69–7.13); this level indicated that the water was acidic. The ammonium level in the water was 17.48 mg/L (normal level is 1–5 mg/L. The water did not contain faecal coliform bacteria. Based on this examination, the water in this area was of low quality [43, 44].

## Anthropometric examination

Maternal anthropometric measurements included height (TB) and body weight (BW) with standard procedures. Each measurement was made two times, and the average of the measurements was taken. The results of these measurements were used to determine body mass index (BMI). Height was measured by placing a microtoise on the wall at two metres from a flat floor. The subject stood upright in the middle of the microtoise without wearing footwear or socks. The subject faced straight ahead, with their shoulders, buttocks and heels were pressed against the wall and their arms free at their sides. The moving part of the microtoise was carefully lowered so that it touched the top of the head, and the hair was pressed. Height measurements were taken twice; the results were read, averaged and recorded on the form [45]. These measurements were taken alternately following health protocols, meaning that the area was cleaned with disinfectant after each subject was measured.

BW was measured by placing a digital scale on a flat and hard floor without a base. Before weighing, the scale was zeroed. Subjects stood upright in the middle of the scale footrest without wearing footwear or socks and wearing minimal clothing. Weight measurements were carried out twice, and the results were averaged [46]. BMI was calculated by dividing the BW in kilograms by the TB squared in metres ($kg/m^2$). The BMI categories for women were as follows: underweight ($<18.5 kg/m^2$), normal weight ($18.5–22.9 kg/m^2$), overweight ($23–24.9 kg/m^2$), obese I ($25–29.0 kg/m^2$), and obese II ($>30 kg/m^2$) [47].

## Anthropometric measurements of children

Anthropometric measurements of BW in children who were able to stand were performed using digital scales, while digital baby scales were used for children who were unable to stand. Height was determined using a microtoise or longboard. Nutritional status was determined based on the WHO classification of malnutrition in children as underweight, stunting and malnutrition or wasting based on BW and TB measurements for children aged 0–23 months and height for children aged 24–60 months [48]. Nutritional status determination was based on the WHO's growth curve, with the underweight category (weight for age) comprising underweight and severe underweight. The category was used if the weight for age is less than 2 standard deviations (SDs) below the mean [48].

The stunting category (height for age) comprised stunting and severe stunting. The category was used if the height for age was less than 2 SDs below the mean. The wasting category (weight for height) comprised wasting and severe wasting, and the category was used if the weight for height was less than 2 SDs below the mean [48].

## Laboratory measurement

Laboratory tests were carried out while following the COVID-19 protocols, with the use of complete personal protective equipment. Laboratory tests were performed by taking maternal blood samples to measure calcium, iron, zinc, vitamin D, pancreatic amylase, and lipase levels. Examination of serum calcium in this study was performed using the Arsenazo III method with ARCHITECTc8000 (Abbott, Illinois, United States), and the reagent material was CAL-CIUM1500T (Abbott). We used a serum sample with stability at 20–25˚C. The results of the serum calcium examination were considered either normal (8.4–10.2 mg/dL) or deficient (<8.4 mg/dL).

The examination of serum zinc levels in this study was performed using inductively coupled plasma-mass spectrophotometry (ICP–MS) with an Agilent 7700 Series ICP–MS (Agilent Technologies, United States). Serum samples with stability at 2–8˚C were used. The results of the serum zinc examination were considered normal at 60–130 mcg/dL or deficient at <60 mcg/dL.

We used ICP–MS to measure serum iron levels in this study using the Agilent 7700 Series ICP–MS. A serum iron level of 35–145 mcg/dL was considered normal, while an iron level <35 mcg/dL was considered to indicate deficiency.

Serum amylase levels were measured using spectrometry with an ARCHITECTc8000 (Abbott, Illinois, United States), and an alpha amylase pancreatic reagent was used (Abbott). Serum samples with a stability of 20–25˚C were used. The results of the examination of pancreatic serum amylase were considered normal at levels of 13–53 U/L, and deficient at levels < 13 U/L.

Serum lipase levels were measured using spectrophotometry with an ARCHITECTc8000, and the Lipase 778T reagent (Abbott) was used. We used serum with a stability of 20–25˚C. Serum lipase levels were considered normal at 13–51 U/L, while deficiency was considered at a level < 13 U/L.

For the examination of vitamin D, namely, serum 25(OH)D levels, the chemiluminescence immune assay (CLIA) method with total Liaison®25-OH Vitamin D (Diasorin, United States) was used. We used serum with a stability of 2–8˚C. The results of the examination of vitamin D levels were considered as follows: <10 ng/mL indicated deficiency, 10–29 ng/ml indicated insufficiency, and 30–100 ng/mL indicated sufficiency.

## Food recall

Our assessment of nutrient intake was based on food recall for two days (one weekday and one weekend day) and included energy, protein, fat, and carbohydrate intake. Calculations were made using the Nutrisurvey 2007 application, which included Indonesian foods [49]. In the 24-hour recall method, subjects and their parents were asked to recall the subject's exact food intake by the nutritionist, who was trained in interviewing techniques. Any two-day assessment interview protocol must be standardized and pretested prior to use. Pretesting was undertaken in an area near the study site, using subjects similar to those who would participate in this study. The nutritionist was retrained to minimize interviewer bias.

Nutrisurvey is a nutrition survey and calculations software application originally developed in Germany with an English translation. This application is useful for nutritional analysis and

calculating energy needs, diet planning, diet history, meal frequency, identifying nutrients in food, and other functions in carrying out nutritional management therapy. This free application can be used to analyse a person's food intake and has been validated.

## Statistical analysis

Data were analysed using the IBM-SPSS v11.5 statistical program (IBM Corp., Chicago, IL, United States). Categorical variables are expressed as percentages. Normally distributed continuous variables are expressed as the mean ± SD, and nonnormally distributed continuous variables are expressed as the median (minimum—maximum). To compare the two groups, the independent T test was used if the distribution was normal, and the Mann—Whitney U test was used for data that were nonnormally distributed. The variables are expressed as *p* values, mean differences±standard errors, and 95% confidence intervals (CIs). All data were stored and properly equipped to prevent data loss. Analysis was carried out after the data were completely collected.

## Ethical consideration

This study was conducted according to the guidelines laid down in the Declaration of Helsinki, and all procedures involving research study participants were approved by the Universitas Sumatera Utara Ethical Committee (No. 423/KEP/USU/2020) and clinicaltrials.gov (NCT04669106) (date of document: November 25th, 2020). Written and verbal informed consent was obtained from all subjects. Verbal consent was witnessed and formally recorded in the statement sheet after explanation.

## Results

This study included 50 mothers with a mean age of 31.06 ± 6.29 years and children aged 2 to 60 months. Mothers were divided into two groups, namely, mothers with children who had normal growth and mothers with children who had stunted growth (Table 1). The group of mothers aged 20–30 years old had the highest proportion of children with stunted, while the 30–40-year age group had a higher proportion of mothers with children with normal growth.

Table 1 shows that there were no significant differences between the groups for any sociodemographic variables. The majority of mothers were housewives (69.5%), although the daily income level showed that a lower percentage earned the stipulated regional minimum wage (mean total income±SD: USD 93.03±57.08) per month. Pregnancy history included the number of children (all subjects: mean = 2.66 ± 1.42 children), age at first pregnancy (all subjects: mean ± SD: 24.3 ± 5.99 years), pregnancy problems (67.8% reported no complaints during pregnancy), history of miscarriage (78% had no history of miscarriage), and history of antenatal care (all subjects: mean ± SD: 6.56 ± 2.0 times), and significant differences were not observed between the two groups.

Table 2 provides data on the anthropometric characteristics and nutritional intake of the mothers. All parameters, including the anthropometric parameters, did not show significant differences between the two groups; however, according to the BMI data, mothers of children with normal growth were more likely to be overweight or obese. Moreover, the mean height of mothers was 154.7 ± 5.62 cm, which is slightly lower than the mean height of 159.8 cm for women in Asia (China). Mothers of children with normal growth were taller than mothers of children with stunted growth, but there was no significant difference in height. There was no significant difference in food intake between the two groups.

Table 3 shows the birth weight, which showed no significant difference between the two groups. It was reported that nine children had low birth weight (<2500 g), and five of them

**Table 1. Maternal sociodemographic data.**

| Variable | Mothers of children with normal growth | Mothers of children with stunted growth | P |
|---|---|---|---|
| Age | | | |
| Mean ± SD (years) | 32.41 ± 5.99 | 28.6 ± 6.25 | 0.05[a] |
| Age category (years): | | | |
| 18–20 | 1 (3.1%) | 1 (5.6%) | 0.34[c] |
| 20–30 | 12 (37.5%) | 11 (61.1%) | |
| 30–40 | 18 (56.3%) | 6 (33.3%) | |
| 40–50 | 1 (3.1%) | 0 | |
| Mother's occupation: | | | |
| Midwife | 1 (3.1%) | 0 | 0.46[c] |
| Teacher | 1 (3.1%) | 1 (5.6%) | |
| Housewife | 24 (75.0%) | 17 (94.4) | |
| Labour | 1 (3.1%) | 0 | |
| Government employee | 5 (15.6) | 0 | |
| Husband's occupation: | | | |
| Labour | 2 (6.3%) | 3 (16.7%) | 0.56[c] |
| Teacher | 2 (6.3%) | 0 | |
| Fisherman | 10 (31.3%) | 5 (27.8%) | |
| Small traders | 17 (53.1%) | 10 (55.6%) | |
| Government employee | 1 (3.1%) | 0 | |
| Mother's education level: | | | |
| No education | 0 | 1 (5.6%) | 0.34[c] |
| Primary School | 13 (40.6%) | 8 (44.4%) | |
| Junior High School | 4 (12.5%) | 3 (16.7%) | |
| Senior High School | 7 (21.9%) | 5 (27.8%) | |
| Bachelor's degree | 8 (25.0%) | 1 (2.0%) | |
| Family income per month: | | | |
| Mean ± SD (USD) | 93.19 ±62.17 | 92.73± 48.49 | 0.98[b] |
| Family income category: | | | |
| ≥RMW (USD 173.69) | 27 (84.4%) | 16 (88.9%) | 1.00[c] |
| ≥RMW (USD 173.69) | 5 (15.6%) | 2 (11.1%) | |
| Number of children: | | | |
| Mean ± SD (count) | 2.56 ± 1.4 | 2.83 ± 1.5 | 0.90[b] |
| Age at first pregnancy: | | | |
| Mean ± SD (years) | 24.38 ± 5.42 | 24.17 ± 7.08 | 0.18[b] |
| Problems during pregnancy: | 0 | 1 (5.6%) | 0.26[c] |
| Fever | 3 (9.4%) | 0 | |
| Abdominal discomfort | 2 (6.3%) | 3 (16.7%) | |
| Nausea and vomiting | 1 (3.1%) | 0 | |
| Abdominal cramps | 26 (81.3%) | 14 (77.8%) | |
| No problems | | | |
| History of miscarriage: | | | |
| Yes | 3 (9.4%) | 1 (5.6%) | 1.0[c] |
| No | 29 (90.6%) | 17 (94.4%) | |
| Antenatal care: | | | |

(*Continued*)

**Table 1.** (Continued)

| Variable | Mothers of children with normal growth | Mothers of children with stunted growth | P |
|---|---|---|---|
| Mean ± SD (times per pregnancy) | 6.84 ± 2.05 | 6.06 ± 1.89 | 0.19[a] |

RMW: Regional minimum wage

[a] Independent t test

[b] Mann—Whitney U test

[c] Fisher's exact test

*significance: $p < 0.05$

were from the group of children with stunted growth. Table 3 also shows the data on history of exclusive breastfeeding. Although there were no significant differences between the two groups, the highest percentage of mothers who exclusively breastfed was in the group who had children with normal growth.

Table 3 also shows the data concerning the nutritional status of the children. Based on weight for age, the two groups of children did not show a significant difference, although it appeared that in the stunting group, 22.2% of the children were underweight or severely underweight, which was higher than the proportion seen in the normal-growth group (9.4%). Based on the stunting category, the stunting group was divided into children who were very short (33.3%) and had short stature (66.7%), whereas in the normal-growth group, most children were considered normal height, although there were also tall children (15.6%).

Table 4 provides the results of measurements of calcium, iron, zinc, vitamin D, pancreatic amylase and lipase levels. These results showed that there was a significant difference in serum calcium levels between the groups of mothers of children with normal and stunted growth and

**Table 2. Maternal anthropometric and nutritional data.**

| Variable | Mothers of children with normal growth | Mothers of children with stunted growth | p |
|---|---|---|---|
| Mothers | | | |
| Body weight | 56.94 ± 11.85 | 57.06 ± 10.29 | 0.84[b] |
| Height | 155.41 ± 6.06 | 153.44 ± 4.60 | 0.21[a] |
| Body mass index | 32.47 ± 4.16 | 24.14 ± 3.93 | 0.58[a] |
| Mother's body mass index category: | | | |
| Underweight | 3 (9.4%) | 1 (5.6%) | 0.89[b] |
| Normal | 15 (46.9%) | 9 (50%) | |
| Overweight | 6 (18.8%) | 2 (11.1%) | |
| Obese 1 | 6 (18.8%) | 5 (27.8%) | |
| Obese 2 | 2 (6.3%) | 1 (5.6%) | |
| Mother's intake per day | | | |
| Calorie intake | 1606.03 ± 298.8 | 1582.5 ± 269.5 | 0.78[a] |
| Carbohydrate intake | 233.29 ± 47.7 | 220.34 ± 42.23 | 0.33[a] |
| Protein intake | 61.28 ± 12.69 | 56.32 ± 14.59 | 0.24[a] |
| Fat intake | 47.68 ± 18.74 | 55.61 ± 25.19 | 0.25[a] |

[a] Independent t test

[b] Mann—Whitney U test

[c] Fisher's exact test

*significance: $p < 0.05$

**Table 3. Children anthropometric and nutritional data.**

| Variable | Children with normal growth | Children with stunted growth | *p* |
|---|---|---|---|
| Age (months) | 28.75 ± 18.4 | 31.89± 21.7 | 0.23[a] |
| Birth weight (g) | 3200 ± 660 | 3100 ± 660 | 0.59 |
| Exclusive breastfeeding | | | |
| Yes | 19 (59.4%) | 9 (50%) | 0.56[c] |
| No | 13 (40.6%) | 9 (50%) | |
| Children growth Category | | | |
| Weight for age: | | | |
| Severe underweight | 2 (6.3%) | 2 (11.1%) | 0.43[c] |
| Underweight | 1 (3.1%) | 2 (11.1%) | |
| Normal | 27 (84.4%) | 14 (77.8%) | |
| Obese | 2 (6.3%) | 0 | |
| Weight for height | | | |
| Severe wasting | 4 (12.5%) | 0 | 0.11[c] |
| Wasting | 2 (6.3%) | 1 (5.6%) | |
| Normal | 24 (75%) | 12 (66.7%) | |
| Obese | 2 (6.3%) | 5 (27.8%) | |
| Child's intake per day | | | |
| Calorie intake | 907.12 ± 351.81 | 791.29 ± 215.82 | 0.16[b] |
| Carbohydrate intake | 207.38 ± 512.05 | 101.41 ± 38.19 | 0.25[b] |
| Protein intake | 34.11 ± 15.19 | 32.38 ± 8.85 | 0.61[a] |
| Fat intake | 31.37 ± 14.99 | 28.93 ± 10.77 | 0.51[a] |

[a] Independent t test

[b] Mann—Whitney U test

[c] Fisher's exact test

*significance: *p* < 0.05

lower serum calcium levels in the group of mothers of children with normal growth (*p* = 0.03, mean difference±standard error (SE) = 0.23±0.12, 95% CI: 0.02–0.45).

Likewise, with the calcium classification, it was seen that 6.3% of mothers in the group of with children with normal growth had calcium deficiency, but there was no significant difference between groups. The parameters of serum iron and zinc levels did not show significant differences between the two groups.

Apart from calcium, serum vitamin D levels did not show any significant difference between the two groups; however, none of the study subjects were categorized as vitamin D deficient (all subjects: mean±SD: 19.0 ± 4.5 ng/mL). However, the levels of calcium and serum 25(OH)D in the group of mothers who had children with stunted growth were higher than those in the group of mothers who had children with normal growth.

With regard to the enzyme examination, serum pancreatic amylase levels did not show any significant differences, but significant differences were seen in serum lipase levels. The mean lipase level in the group of mothers who had children with stunted growth was significantly lower than that in the group of mothers who had children with normal growth (*p* = 0.02, median = 4.34, 95% CI: 0.62–8.06). There was no significant correlation between calcium and lipase (*p* = 0.51) or between calcium and vitamin D (*p* = 0.067) levels.

**Table 4. Differences in the nutritional data.**

| Variable | Mothers of children with normal growth | Mothers of children with stunted growth | P |
|---|---|---|---|
| Calcium serum level (mg/dL) | 8.82 ± 0.35 | 9.05 ± 0.37 | 0.03*[b] |
| Calcium category: | | | |
| Deficiency | 2 (6.3%) | 0(0%) | 0.53[c] |
| Normal | 30 (93.8%) | 18 (100%) | |
| Serum iron level (mcg/dL) | 90.25 ± 44.38 | 94.72 ± 28.91 | 0.67[a] |
| Iron category: | | | |
| Low | 1 (3.1%) | 0(0%) | 0.54[c] |
| Normal | 27 (84.4%) | 17 (94.4%) | |
| High | 4 (12.5%) | 1 (5.6%) | |
| Serum zinc level (mcg/dL) | 71.84 ± 7.97 | 75.0 ± 9.37 | 0.24[a] |
| Zinc category: | | | |
| Low | 2 (6.3%) | 1 (5.6%) | 1.0[c] |
| Normal | 30 (93.8%) | 17 (94.4%) | |
| Serum vitamin D level (ng/mL) | 19.0 ±4.51 | 20.1 ± 5.14 | 0.45[a] |
| Vitamin D category: | | | |
| Deficiency | 0 (0%) | 1 (5.6%) | 0.36[c] |
| Insufficiency | 32 (100%) | 17 (94.4%) | |
| Sufficiency | 0 (0%) | 0 (0%) | |
| Amylase pancreatic (U/L) | 26.03 ± 5.11 | 25.44 ± 8.03 | 0.78[a] |
| Amylase category: | | | |
| Low | 0 (0%) | 0 (0%) | - |
| Normal | 32 (100%) | 18 (100%) | |
| Lipase (U/L) | 20.06 ± 6.61 | 15.72 ± 6.01 | 0.03*[a] |
| Lipase category: | | | |
| Low | 4 (12.5%) | 7 (38.9%) | 0.04*[c] |
| Normal | 28 (87.5%) | 11 (61.1%) | |

[a] Independent t test

[b] Mann—Whitney U test

[c] Fisher's exact test

*significance: $p < 0.05$

## Discussion

We investigated factors associated with stunting in 18 children who exhibited stunted growth in a village North Sumatra, Indonesia. A stunting prevention program has been carried out; however, a reanalysis of why cases of stunting are still occurring, despite there being decreased stunting rates in the surrounding area, is required. There are various causes of stunting, such as maternal nutritional status, anaemia, inadequate weight gain during pregnancy, short pregnancy duration, and a very young age at first pregnancy [1, 50–54]. In addition to these factors, there are other fundamental factors, such as low maternal education levels, poor sanitation, and not being exclusively breastfed [55–58]. In our study, there were no differences in sociodemographic characteristics (education level, family income, etc.) among children with normal growth and stunted growth. For this reason, it is necessary to examine the underlying factors of the incidence of stunting in toddlers.

Our study focused on the nutritional status of mothers who have children under five years old. To examine maternal nutritional status, we measured calcium, iron, and zinc levels,

deficiencies of which are associated with growth problems and anaemia. An important vitamin related to metabolism is vitamin D, and its measurement requires an understanding of the interaction between calcium and vitamin D. Moreover, the function of the maternal digestive system is related to metabolism and vitamins, so disturbances that occur can affect mineral and vitamin activity.

Stunting is a disease related to long-term nutritional deficiency, especially during the first 1000 days of a child's life [59–63]. This nutritional deficiency is caused by a lack of macronutrients and micronutrients [64, 65]. Macronutrient diseases are caused by a deficiency of protein, fat and carbohydrates [63, 64]. Children with these deficiencies usually show clinical symptoms that directly affect their nutritional status [64, 66–68]. However, micronutrient deficiencies of minerals and vitamins are often not detected because patients initially present with subclinical symptoms and have long-term defects [69–72]. These defects lead to impaired nutritional status in children and can have serious implications when in adulthood [63].

If babies with stunted growth can catch up and grow quickly, the stunting rate can be reduced and the stunting cycle can be broken between generations. Although most babies with stunting at an early age catch up in growth after the age of two years and during adolescence, there are some metabolic disorders that can occur in adulthood [63, 73, 74]. As a result of the pursuit for growth in height after a delay in early growth, the growth of other organs may be stunted, and cognitive development is also affected, which is linked to obesity in adulthood and associated with diabetes and heart disease [60–63, 75].

Deficiencies in calcium and vitamin D are related to bone growth inhibition, but the metabolism of calcium and vitamin D is closely related to absorption in the body, including the role of the body's digestive enzymes [33, 76–78]. Our study found that there was a significant difference in serum calcium levels between the two groups; mothers of children with stunted growth had significantly higher calcium levels than mothers of children with normal growth, even though both groups were classified as having calcium levels within the normal range. Serum lipase levels were lower in mothers of children with stunted growth. While our study showed that serum 25(OH)D representing circulating vitamin D were not normal, most cases involved vitamin D insufficiency and deficiency, although no correlation was found between any of these minerals and vitamins, which may be related to the small sample size.

Stunting is a result of long-term nutrient deficiency. All types of nutrient deficiencies can be related. Calcium and vitamin D are two nutrients that affect bone mineral density during pregnancy and a child's development [70–72, 79, 80]. Maternal nutritional status greatly affects the nutritional status of children during the first 1000 days of life, including the availability of maternal calcium, which will affect foetuses' calcium levels during pregnancy [63, 81, 82]. Our study found that mothers who had children with stunted growth had high serum calcium levels, low serum vitamin D levels, and low serum lipase levels. Mothers of children with normal growth had low serum calcium levels but were still classified as normal, low serum vitamin D levels, and significantly different serum lipase levels.

During pregnancy, the mother provide calcium to the foetus for growth and development [72, 76, 83–85]. During this period, there are major changes in the body to meet calcium needs and maintain maternal homeostasis [69, 79, 86, 87]. Calcium homeostasis is a complex process involving calcium and parathyroid hormone, calcitonin, and 1,25-dihydroxyvitamin D3 [70–72]. The total serum calcium concentration decreases during pregnancy, and there is increased demand for haemodilution along with a decrease in albumin levels [72]. However, constant serum calcium levels will be maintained through the homeostasis control mechanism so that the body's calcium needs can be met [83–85, 88, 89].

Homeostasis affects the absorption of calcium, and during pregnancy, there is an increase in intestinal calcium absorption [76, 83, 90]. Calcium absorption is also influenced by fat [76–

78, 90, 91]. Fat intake can stimulate calcium absorption depending on the type and amount of fat [77]. Previous research has suggested that a high-fat diet may aid calcium absorption in human and nonhuman animal studies [78]. A high intake of saturated fat causes hyperpermeability of the intestinal membrane, which increases the passive entry of calcium [76, 77].

Calcium ions play an important role in calcium absorption, affecting the rate of lipid hydrolysis [77]. The digestion of fat from long-chain fatty acids is more difficult if there is no support from lipase as a fat-breaking enzyme [77, 91]. Calcium precipitates the accumulated free fatty acids, thereby releasing them from the surface of the lipid granules and making it easier for the lipases to emulsify the lipids [78, 90, 91]. Calcium ions can increase the rate of lipolysis through this mechanism [76]. The deposits formed between calcium and long-chain saturated fatty acids are more difficult to absorb, thus decreasing lipid bioavailability [76, 90].

If the levels of lipase produced by the liver are increased in the blood, then the number of osteoblasts decreases, and the number of osteoclasts increases [33, 86]. Osteoclasts are needed for bone resorption in pregnant women, which contributes to meeting foetal calcium needs [86, 92]. This mechanism is also influenced by parathyroid hormone-related peptide (PTHrP), which increases bone resorption in pregnant women [70, 86]. This can be seen in the increase in PTHrP levels in breast milk [69–72, 86, 93]. In our study, serum lipase levels were significantly decreased in the group of mothers of children with stunted growth, which would lead to an increase in the number of osteoblasts and a decrease in the number of osteoclasts. As a result of this decrease in the number of osteoclasts, the mother would have difficulty meeting the foetus's calcium needs because the calcium in the mother's bones cannot be degraded; therefore, the foetus would be born with a low calcium state, causing stunting. This mechanism could be proposed to understand the role of lipase in the occurrence of stunting.

In this study, the serum lipase levels of mothers of children with normal growth were higher than those of mothers of children with stunted growth. The possible mechanism is that high lipase levels can help fat absorption and increase the absorption of calcium from food before pregnancy and during pregnancy. If a mother's serum lipase level is low, then optimal fat absorption will not occur, which will ultimately inhibit calcium absorption. The possible mechanism explaining stunted growth in children is that their mothers had low lipase levels and were unable to meet the foetal needs for calcium due to a decreased number of osteoclasts, even though the serum calcium levels were normal. High lipase levels lead to high fat absorption, which corresponds to the absorption rate of calcium within the foetal body; therefore, in the event of low lipase levels in mothers, obstructed calcium absorption could lead to stunted growth in children. This could be coupled with the suboptimal absorption of calcium due to low serum vitamin D levels.

The nutritional status of a women is important in every stage of her reproductive period, not only prenatally, including not only her body mass index or height but also her vitamin/ mineral and enzyme status after giving birth. This can describe the condition of mothers prenatally and during pregnancy. However, the status of vitamins, minerals and enzymes in mothers deserves attention because the possibility of a mother becoming pregnant again is very high. Moreover, mothers may not pay attention to the interval between the birth of one child and another. This condition is one of the reasons why examination of vitamin/mineral and enzyme status is a main focus and is carried out after birth.

Limitations in this study include the small sample size and the lack of significance of our results (mostly); these differences might be considered a result of normal individual variation rather than group differences. We also did not test the mothers when they were pregnant, so their lipase levels might have changed in the period between birth and our study. Another limitation is that there were no data on maternal parathyroid hormone, calcium, and vitamin D levels and lipase levels in children. Our findings can form the basis for further research,

although this study was limited by the small sample size, single study location, and the lack of analysis of the surrounding soil and climate factors. There are still many factors that must be studied to establish the mechanism of calcium regulation in mothers who have children with stunted growth. It is hoped that this research can be generalized to a wider population based on the research methodology used, including the sampling methods and data collection, which were carried out correctly.

## Conclusions

Serum lipase levels were significantly lower in the group of mothers of children with stunted growth when compared with the group of mothers of children with normal growth in one village in North Sumatra, Indonesia. A low serum lipase level probably results in a mother being unable to meet the calcium needs of her child during pregnancy, increasing the child's risk of stunted growth. Serum calcium levels were significantly higher in the group of mothers of children with stunted growth, possibly due to the interaction among calcium, lipase and vitamin D, which was found to be low in all study subjects. It is hoped that the results of this study will provide further understanding of the role of lipase in calcium metabolism so that programs for the early prevention of growth stunting can be implemented for pregnant women.

## Supporting information

**S1 Table. Maternal sociodemographic data.**
(PDF)

**S2 Table. Maternal anthropometric data.**
(PDF)

**S3 Table. Children anthropometric and nutritional data.**
(PDF)

**S4 Table. Differences in the nutritional data.**
(PDF)

**S1 Dataset. Data set.**
(SAV)

**S1 Data. Water examination data.**
(PDF)

**S2 Data. Soil examination data.**
(PDF)

**S1 Checklist. Strobe check.**
(PDF)

## Author Contributions

**Conceptualization:** Dina Keumala Sari, Kraichat Tantrakarnapa.

**Formal analysis:** Dina Keumala Sari, Rina Amelia.

**Funding acquisition:** Dina Keumala Sari.

**Investigation:** Rina Amelia.

**Methodology:** Dina Keumala Sari, Rina Amelia, Dewi Masyithah, Kraichat Tantrakarnapa.

**Resources:** Dina Keumala Sari, Dewi Masyithah, Kraichat Tantrakarnapa.

**Software:** Rina Amelia.

**Supervision:** Dina Keumala Sari, Kraichat Tantrakarnapa.

**Validation:** Dewi Masyithah.

**Visualization:** Dewi Masyithah.

**Writing – original draft:** Dina Keumala Sari.

**Writing – review & editing:** Rina Amelia, Dewi Masyithah.

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
