## [Decision Letter · Decision Letter 0]

26 Jan 2022

PONE-D-21-01488

EVALUATING THE EFFECT OF LOW LIPASE ENZYME SERUM ON STUNTING PROBLEMS IN NORTH SUMATRA, INDONESIA

PLOS ONE

Dear Dr. Sari,

Thank you for submitting your manuscript to PLOS ONE. After careful consideration, we feel that it has merit but does not fully meet PLOS ONE’s publication criteria as it currently stands. Therefore, we invite you to submit a revised version of the manuscript that addresses the points raised during the review process.

The manuscript has been evaluated by two reviewers, and their comments are available below.

The reviewers have raised a number of minor concerns. They feel that the title should be modified to better reflect the results, and the discussion should be more concise. The reviewers also note some typographical errors that need attention. 

Could you please carefully revise the manuscript to address all comments raised?

<o:p></o:p>

We look forward to receiving your revised manuscript.

Kind regards,

Lorena Verduci

Staff Editor

PLOS ONE

Journal Requirements:

Reviewers' comments:

Reviewer's Responses to Questions

**Comments to the Author**

1. Is the manuscript technically sound, and do the data support the conclusions?

Reviewer #1: Yes

Reviewer #2: Yes

2. Has the statistical analysis been performed appropriately and rigorously? 

Reviewer #1: Yes

Reviewer #2: Yes

3. Have the authors made all data underlying the findings in their manuscript fully available?

Reviewer #1: No

Reviewer #2: Yes

4. Is the manuscript presented in an intelligible fashion and written in standard English?

Reviewer #1: Yes

Reviewer #2: Yes

5. Review Comments to the Author

Reviewer #1: General Comments: This cross-sectional study assessed the “effect” of low lipase enzyme serum on stunting problems in North Sumatra, Indonesia. Although the manuscript is generally well written, it could benefit from revision.

Revisions being requested

Title: Cross-sectional study design does not evaluate the “effect” of something because data on both exposure and outcome are collected at the same time and there is no way to determine if exposure led to an outcome.

The word “problems” is not relevant in the title.

Abstract

1. Lines 33 - 35: Grammatical error, revise the sentence “The objectives …. “.

Introduction2

2. Lines 78 - 79: The statement “To the best … “ is misleading. How does this statement agree with lines 69 - 70? Do you mean a study like yours was not conducted in the study area before?

Materials and Methods

3. Lines 94 and 123: The sections “study design and participants” and “participants” should be reorganised.

4. Lines 116 – 121: Please move these words under the sub-section “Ethical considerations” which should be the last sub-section of “Materials and methods”.

5. Lines 136 - 137: You stated that soil and water content were checked in a laboratory but I did not see any results on these.

6. Lines 140 – 147: There is no need to capitalize “microtoise”.

7. Lines 166 – 169: Make reference to anthropometric z-scores in the definition of wasting, stunting and underweight.

8. Line 214: Insert “and” between error and 95% confidence interval.

9. Table 1: Add unit of measurement to category of age.

10. Table 1: Last education, Define SD, SMP, SMA and S1.

11. Table 1: Family income per month, give US dollar equivalent of amount beneath the table.

12. Lines 242 - 243: Grammatical error, revise sentence “The group …”.

13. Table 2: Children growth category, I do not see rationale for including height-for-age under this label as the children were divided into two groups using height-for-age z-score.

Discussion

14. Line 331: Remove the “of” at the end of the line.

15. Line 377: Mechanism is incorrectly spelt.

16. Lines 385 - 243: Grammatical error, revise sentence “We also …”.

17. Lines 390 - 392: The word “need” is not necessary in that sentence.

Reviewer #2: The manuscript is well written in clear English, however some typographical errors that need attention are highlighted below:

Line 113: …”height per age”… this could read better as “ …height for age…”

Line 119: How did the authors decide on who to use verbal consent and who to use written consent? What rationale was used to decide?

Line 128: What could have been the result of recruiting clients for the study as described? Did the authors consider the risk of selection bias? How was this minimized or mitigated?

Line 129: Was enrolment done before applying the exclusion criteria?

Line 130-131: What was the rationale for dividing the subjects into the two groups?

Line 242: …were taller be higher than… : correct the typos

Line 263: try and keep discussion out of the results section.

Table 3: For vitamin D serum level, how did the authors define “Deficiency” and “Insufficiency” in this study?

Line 291: …..been completely work in this area….: needs correction.

Line 377: “mechanisms” should have no “s” and instead of understand could authors could use another word for clarity.

Line 385-386: repetition of words should be corrected and revise sentence for clarity.

Results:

The presentation of results does not clearly bring out what is stated in the title: “Evaluating the effect of low lipase enzyme serum on stunting problems in North Sumatra, Indonesia”. There seems to be some disconnect between the two.

Discussion:

The discussion is rather long which leads to a tendency to include a lot of information.

6. PLOS authors have the option to publish the peer review history of their article (what does this mean?). If published, this will include your full peer review and any attached files.

Reviewer #1: No

Reviewer #2: No

---

## [Author Response · Author response to Decision Letter 0]

3 Feb 2022

RESPONSE TO REVIEWERS

Comments :The reviewers have raised a number of minor concerns. They feel that the title should be modified to better reflect the results, and the discussion should be more concise. The reviewers also note some typographical errors that need attention. 

Revised :Yes, I already changed the title: Low Levels of Lipase Serum In Mothers Who Have Stunting Children Provide The Possibility of Low Calcium Absorption During Pregnancy in North Sumatra, Indonesia (in line 4-6).

Comment

Revisions being requested Title: Cross-sectional study design does not evaluate the “effect” of something because data on both exposure and outcome are collected at the same time and there is no way to determine if exposure led to an outcome.

The word “problems” is not relevant in the title.

Revised :Yes, I already revised it to title that showed the results of the study

Revision:

Abstract

1. Lines 33 - 35: Grammatical error, revise the sentence “The objectives …. “.

Revised: 

This study aimed to investigate the differences in sociodemographic factors, mineral, vitamin, and enzymes parameters associated with the occurrence of stunting in mothers with normal and stunting children. (line 34-36)

Introduction

2. Lines 78 - 79: The statement “To the best … “ is misleading. How does this statement agree with lines 69 - 70? Do you mean a study like yours was not conducted in the study area before?

Revised:

I already change it with this statement: “Based on previous research, there is still more needed to explore the role of vitamins, minerals, and enzymes to prevent stunting” (line 79-80)

I do agree, I hope it will not mislead the goals of this study and the reader will not misinterpretation about this study.

3. Materials and Methods

Lines 94 and 123: The sections “study design and participants” and “participants” should be reorganised.

Revised:

I already revised it, to reorganized the section, please check in line 95 and 118.

4. Lines 116 – 121: Please move these words under the sub-section “Ethical considerations” which should be the last sub-section of “Materials and methods”.

Revised:

I already revised it, please check in line 215.

5. Lines 136 - 137: You stated that soil and water content were checked in a laboratory but I did not see any results on these.

Revised:

Yes, I already add the information about the soil and water, please check in line 133-143

6. Lines 140 – 147: There is no need to capitalize “microtoise”.

Revised: 

Yes, I already revised it, please check 146-156.

7. Lines 166 – 169: Make reference to anthropometric z-scores in the definition of wasting, stunting and underweight.

Revised:

Yes, I already revised it, please check in line 172-179.

8. Line 214: Insert “and” between error and 95% confidence interval.

Revised:

Yes, I already revised it, please check in line 224

9. Table 1: Add unit of measurement to category of age.

Revised:

Yes, I already revised it in table 1

10. Table 1: Last education, Define SD, SMP, SMA and S1.

Revised:

Yes, I already revised it in table 1

11. Table 1: Family income per month, give US dollar equivalent of amount beneath the table.

Revised:

Yes, I already revised them. I also revise in line 252.

12. Lines 242 - 243: Grammatical error, revise sentence “The group …”.

Revised:

Yes, I already revised in line 261

13. Table 2: Children growth category, I do not see rationale for including height-for-age under this label as the children were divided into two groups using height-for-age z-score.

Revised:

Yes I already revised it in the table 2.

Discussion

14. Line 331: Remove the “of” at the end of the line

Revise:

Yes, I already revised it in line 347-348. I already delete ‘of”.

15. Line 377: Mechanism is incorrectly spelt.

Revise:

Yes, I already revised it in line 393

16. Lines 385 - 243: Grammatical error, revise sentence “We also …”.

Revise:

Yes, please check in line 359: “Mothers with normal growth children” to “mothers with normal children group”

Yes please check in line 401: “We also did not you did not test mothers” to “we did not test the mothers.

17. Lines 390 - 392: The word “need” is not necessary in that sentence.

Revise:

Yes, I already revised it. Line 406

Reviewer #2: The manuscript is well written in clear English, however some typographical errors that need attention are highlighted below:

Line 113: …”height per age”… this could read better as “ …height for age…”

Revised: yes, thank you, I already revised it to all the manuscript.

Line 119: How did the authors decide on who to use verbal consent and who to use written consent? What rationale was used to decide?

Comments: we used for written and verbal informed consent in this study to all the subject. We did not separate or choose which subject use the verbal and written. We explained all the protocol to the subject, and we ask for agreement from the subject verbally but we also asked subject to read and signed the approval after explanation letter.

Line 128: What could have been the result of recruiting clients for the study as described? Did the authors consider the risk of selection bias? How was this minimized or mitigated?

Comments:

Yes, in this study we selected subjects according to the inclusion criteria, namely mothers of a certain age who have stunted children and mothers who have normal children. This will introduce bias, but we will encounter difficulties if we use a random sampling system. this is because the number of participation of mothers with stunting children is different from that of mothers in the normal group. This selection bias cannot be controlled, it can only be prevented, but we did try to improve the quality of examinations and interviews properly and correctly so that the results of this study can be accounted for and the results of the sample can describe the results in a population.

Line 129: Was enrolment done before applying the exclusion criteria?

Comments:

Yes, it is true that registration is done before applying the exclusion criteria, inclusion criteria are carried out first and then exclusion criteria are applied. Therefore, after passing the exclusion criteria, there was a decrease in the number of samples. a total of twelve samples were excluded for several reasons.

Line 130-131: What was the rationale for dividing the subjects into the two groups?

Comments:

the reason to dividint the subjects into the two groups is to see the strength of the causal relationship. It is hoped that with good planning, careful implementation, and proper analysis, this study can make a meaningful contribution.

Line 242: …were taller be higher than… : correct the typos

Revised:

Yes, I already revised it please check in line 261.

Line 263: try and keep discussion out of the results section.

Revised:

I already revised it by deleting the sentences that should in discussion. In line 281-283.

“This result was interesting because there was a significant difference in serum calcium levels in the group of mothers who have normal and stunted children, but it appeared that low serum calcium levels were in the group of mothers with normal children (p = 0.03, mean difference�standard error (SE)= 0.23�0.12, 95% CI: 0.02–0.45)”.

Table 3: For vitamin D serum level, how did the authors define “Deficiency” and “Insufficiency” in this study?

Comments:

Yes, thank you. In line 207-209 in Methods section, in laboratory measurement, there were already state that: “The results of the examination of vitamin D levels were considered deficient in the range of serum 25 (OH) D levels, namely <10 ng/mL, while the insufficient category was 10–29 ng/ml, and the sufficient category was 30–100 ng/mL”.

Line 291: …..been completely work in this area….: needs correction.

Revised:

Yes I already revised it in line 313-314. 

Line 377: “mechanisms” should have no “s” and instead of understand could authors could use another word for clarity.

Comments:Yes, I already revise it, it is also mean for this explanation can make the reader to understand about the role of lipase.

Line 385-386: repetition of words should be corrected and revise sentence for clarity.

Revised: 

I already revised it, please check in line: 408.

Results:

The presentation of results does not clearly bring out what is stated in the title: “Evaluating the effect of low lipase enzyme serum on stunting problems in North Sumatra, Indonesia”. There seems to be some disconnect between the two.

Comments:

Yes I already revised the tittle with:

Low Levels of Lipase Serum In Mothers Who Have Stunting Children Provide The Possibility of Low Calcium Absorption During Pregnancy in North Sumatra, Indonesia

Discussion:

The discussion is rather long which leads to a tendency to include a lot of information.

Comments:

Yes, already revised it, we already deleted in paragraph 2 to reduce the discussions.

---

## [Decision Letter · Decision Letter 1]

21 Sep 2023

PONE-D-21-01488R1Low Levels of Lipase Serum In Mothers Who Have Stunting Children Provide The Possibility of Low Calcium Absorption During Pregnancy in North Sumatra, IndonesiaPLOS ONE

Dear Dr. Sari,

Thank you for submitting your manuscript to PLOS ONE. After careful consideration, we feel that it has merit but does not fully meet PLOS ONE’s publication criteria as it currently stands. Therefore, we invite you to submit a revised version of the manuscript that addresses the points raised during the review process.

Unfortunately, the previous reviewers were unavailable, so additional reviewers were consulted in this round. One of these reviewers has raised a number of concerns that need attention. They request additional information on methodological aspects of the study, and they question the validity of the conclusions given the limitations of the study.

Could you please revise the manuscript to carefully address the concerns raised? As part of your revision, please fill out and upload a copy of the STROBE checklist (https://www.strobe-statement.org/checklists/).

We look forward to receiving your revised manuscript.

Kind regards,

Marianne Clemence

Staff Editor

PLOS ONE

Reviewers' comments:

Reviewer's Responses to Questions

**Comments to the Author**

1. If the authors have adequately addressed your comments raised in a previous round of review and you feel that this manuscript is now acceptable for publication, you may indicate that here to bypass the “Comments to the Author” section, enter your conflict of interest statement in the “Confidential to Editor” section, and submit your "Accept" recommendation.

Reviewer #3: All comments have been addressed

Reviewer #4: (No Response)

2. Is the manuscript technically sound, and do the data support the conclusions?

Reviewer #3: Yes

Reviewer #4: No

3. Has the statistical analysis been performed appropriately and rigorously? 

Reviewer #3: Yes

Reviewer #4: I Don't Know

4. Have the authors made all data underlying the findings in their manuscript fully available?

Reviewer #3: Yes

Reviewer #4: No

5. Is the manuscript presented in an intelligible fashion and written in standard English?

Reviewer #3: Yes

Reviewer #4: Yes

6. Review Comments to the Author

Reviewer #3: from my evaluation, the previous reviewers have raised a number of minor concerns. They feel that the

title should be modified to better reflect the results, and the discussion should be more

concise. The author has done all the needed corrections.

Reviewer #4: EVALUATING THE EFFECT OF LOW LIPASE ENZYME SERUM ON 5 STUNTING PROBLEMS IN NORTH SUMATRA, INDONESIA

-Consider revising the title. I think it does not capture the entirety of the whole article. The use of effect may not be suitable since the study is cross-sectional.

-The journal article needs language editing.

-The introduction did not provide substantive reasons of why mother’s serum lipase enzyme is the focus of the article.

Methodology

-Did you collect low birth weight data? I think this variable is important in your research.

-page 8, line 200-205 – Consider adding the validation procedures.

- Are your subjects randomly selected? If not, can your data be analyzed using standard statistical test?

Results

-Table 1 – Add information on US$ for income

-Table 2 – Consider using BMI instead of Weight for height for mother’s nutritional status.

-Table 2 – Consider separating the data for mother and children.

-Table 3 – Consider separating the results of serum mineral level from mineral status by a secondary line.

Discussion

-Did you use previous articles on post-natal effect on stunting when you drafted the proposal for this research? Is this research output a part of a bigger research?

-The age group of children is from 2-60 months. Did you analyze the data between mothers of less than 2 years old (breastfeeding) and those 2 years old and above. In addition, breastfeeding mothers may have different mechanism than non-breastfeeding mothers.

-What are the confounders in this research?

-There were many ‘probably’ in the article. Consider expounding the biological mechanism of lipase given that calcium is higher among mother of non-stunted children.

- Lines 354-378 the mechanism explained is for pregnant mothers (which is not the subject of your study). Can you find a more relevant mechanism for post-natal women?

Line 386-387 - “We also did not you did not test mothers when they were carrying babies, so it might their lipase levels have changed in the period intervening birth and our study. Another limitation is that there were no data on the maternal parathyroid hormone, calcium, vitamin D, and lipase levels in children.” Do you think your explanations and conclusion are valid given these limitations?

7. PLOS authors have the option to publish the peer review history of their article (what does this mean?). If published, this will include your full peer review and any attached files.

Reviewer #3: No

Reviewer #4: No

---

## [Author Response · Author response to Decision Letter 1]

17 Oct 2023

EVALUATING THE EFFECT OF LOW LIPASE ENZYME SERUM ON STUNTING PROBLEMS IN NORTH SUMATRA, INDONESIA

the revised title: 

Low serum lipase levels in mothers of children with stunted growth indicate the possibility of low calcium absorption during pregnancy: a cross-sectional study in North Sumatra, Indonesia

-Consider revising the title. I think it does not capture the entirety of the whole article. The use of effect may not be suitable since the study is cross-sectional.

Response:

The title already revised with the title that better reflection to the results. 

The title also add: A cross sectional study to let the other researcher can capture the entirety of the whole article by seing the title.

Please check in the line 1-3

-The journal article needs language editing.

Response:

Yes, I already upload the English editing summary, with American Journal Editing service

-The introduction did not provide substantive reasons of why mother’s serum lipase enzyme is the focus of the article.

Response:

Yes, I already add statement about this substantive reasons. Based on previous research there is lack of studies about enzyme that related to nutrition absorption, and in this research we focus on two enzymes which are amylase and lipase. We also interest with lipase because of lipase affect fat absorption that will also affect calcium absorption.

Please check in the line 356-358 and follow on the line 358-362.

Methodology

-Did you collect low birth weight data? I think this variable is important in your research.

Response:

Yes, We collect these data. We presented in the table 3, row 2. We also add analysis in the line 403-405. Probably because of the stunting children only 18 children, compare with normal children 32, so this could results no significance difference. This is from our view about this study, the probable answer to this non-significance results.

-page 8, line 200-205 – Consider adding the validation procedures. 

Response:

Yes, we already add this validation information. NutriSurvey application already. 

Please check on the line 184-188. 

- Are your subjects randomly selected? If not, can your data be analyzed using standard statistical test?

Response:

Yes it was randomly selected. I already add explanation about sample selected process. Please check in the line 628-632.

Results

-Table 1 – Add information on US$ for income

Response:

Yes please check in the table 1 page 14-15, we already revise it

-Table 2 – Consider using BMI instead of Weight for height for mother’s nutritional status. 

Response:

Yes we already revised it, please check in table 2, page 16-17, we already revise it

-Table 2 – Consider separating the data for mother and children.

Response:

Yes, we already revised it in table 2 and 3, already separate, page 16-17 (table 2) and 17-18 (table 3).

-Table 3 – Consider separating the results of serum mineral level from mineral status by a secondary line.

Response:

Yes, we already revise it in table 4. In the page 19. Secondary line and we put space to make the readers of this article easier to understand the table.

Discussion

-Did you use previous articles on post-natal effect on stunting when you drafted the proposal for this research? 

Response:

Most of the previous article focusing in pre-natal effect on stunting, we lack of studies on post-natal effect, but, we still focusing in mother’s nutrition status post-natal effect. The nutritional status of the mother is important, not only pre-natally including observing the mother's body mass index or height, but also the vitamin-mineral and enzyme status of the productive mother after giving birth. This can describe the condition of the mother pre-natally and during pregnancy.

However, this is a deeper concern because we should pay attention to the status of vitamins, minerals and enzymes because the possibility of a productive mother becoming pregnant again is very large. Moreover, mothers think that they do not pay attention to the birth distance between one child and another. This condition is one of the reasons why examination of vitamin-mineral and enzyme status is the main focus and is also carried out after birth. 

Is this research output a part of a bigger research? 

Response:

No, this is an independent study with the hypothesis that to find the difference of vitamin-mineral and enzyme between stunting and normal group in post natal mother. After this publication, we will continue to find the exact mechanism and the different methodology to find this relationships or effect lipase on stunting. This is a base data to expand the next research. If we can find more results, this will help to understand stunting problem and reduce stunting rates in Indonesia. We will move to the next steps, with another aims and hypothesis.

-The age group of children is from 2-60 months. Did you analyze the data between mothers of less than 2 years old (breastfeeding) and those 2 years old and above. In addition, breastfeeding mothers may have different mechanism than non-breastfeeding mothers. 

Response:

Yes we will analyze it, but we still consider for sample size, we need bigger sample size to have normal distribution data and we can find the mechanism for non-breastfeed mother, but for this study, still we lack of information about this breastfeeding mechanism. We do hope for the next research after this publication.

-What are the confounders in this research?

Response:

Confounding factors in this study are mothers with gastrointestinal disorders and mothers with metabolic disorders, we control it in inclusion and exclusion criterias. Please see the line 626-627.

-There were many ‘probably’ in the article. Consider expounding the biological mechanism of lipase given that calcium is higher among mother of non-stunted children. 

Response:

Yes, we already explain the mechanism, we hope it will open other research about this. Please check in line 2065-2075.

- Lines 354-378 the mechanism explained is for pregnant mothers (which is not the subject of your study). Can you find a more relevant mechanism for post-natal women? 

Response:

Yes, we discuss about this that it is very important to have normal status nutrition for every productive mother, along every stage of her productive period, not only pre-natally including observing the mother's body mass index or height, but also the vitamin-mineral and enzyme status of the productive mother after giving birth. This can describe the condition of the mother pre-natally and during pregnancy. However, this is a deeper concern because we should pay attention to the status of vitamins, minerals and enzymes because the possibility of a productive mother becoming pregnant again is very large. Moreover, mothers think that they do not pay attention to the birth distance between one child and another. This condition is one of the reasons why examination of vitamin-mineral and enzyme status is the main focus and is also carried out after birth. 

We already put this analysis in page 2076-2196.

Line 386-387 - “We also did not you did not test mothers when they were carrying babies, so it might their lipase levels have changed in the period intervening birth and our study. Another limitation is that there were no data on the maternal parathyroid hormone, calcium, vitamin D, and lipase levels in children.” Do you think your explanations and conclusion are valid given these limitations? 

Response:

In this study we present the results of the research using the correct methodology, it is hoped that we can provide valid results. as long as these results pave the way for further research, it is hoped that new findings will be obtained to overcome stunting, especially in Indonesia. The prevalence of stunting has decreased in accordance with government programs, but it still exists and is growing. It is hoped that this research can open up other research to examine the causes. Over the last 10 years, improvements in nutritional status have been implemented, but it is possible that there are other causes that can be explored. Something new and understandable if we get results that need to be developed.

---

## [Editor Report · Decision Letter 2]

23 Jan 2024

Low serum lipase levels in mother of children with stunted growth indicate the possibility of low calcium absorption during pregnancy: a cross-sectional study in North Sumatra, Indonesia

PONE-D-21-01488R2

Dear Dr. Dina Keumala Sari,

We’re pleased to inform you that your manuscript has been judged scientifically suitable for publication and will be formally accepted for publication once it meets all outstanding technical requirements.

Kind regards,

Dhruba Shrestha, MD

Academic Editor

PLOS ONE
---

## [Editor Report · Acceptance letter]

14 May 2024

PONE-D-21-01488R2 

PLOS ONE

Dear Dr. Sari, 

I'm pleased to inform you that your manuscript has been deemed suitable for publication in PLOS ONE. Congratulations! Your manuscript is now being handed over to our production team.

Kind regards, 

on behalf of

Dr. Dhruba Shrestha 

Academic Editor

PLOS ONE